# Diels-Alder reaction affords circumpyrene tetracarboxydiimide with excited state intramolecular charge transfer character
Qiang Chen [1,2,3], Michele Guizzardi[2,4], Francisco Méndez [1,5] ✉, Chengwei Ju [1], Helena R. Keller [4], Giulio Cerullo [4], Silvio Osella[6], Francesco Scotognella[4,9], Giuseppe M. Paternò [4] ✉, Klaus Müllen[1,7] & Akimitsu Narita [1,8] ✉

Large polycyclic aromatic hydrocarbon (PAH) imides are promising candidates for optoelectronic applications in view of their narrow optical gaps and/or excited state charge transfer character. Diels–Alder (D–A) reactions of PAHs at bay regions can enable simultaneous extension of the aromatic structure and introduction of the imide moieties, but the actual use of this strategy has been limited. Herein, we demonstrate the D–A cycloaddition of dibenzo[*hi,st*]ovalene, one of the largest PAHs functioning as bisdiene, with maleimides to afford circumpyrene tetracarboxydiimides. Notably, efforts to optimize the yield of di-adduct revealed that the fully aromatized mono-adduct is inert toward further D–A reaction, and density functional theory (DFT) calculations instead indicated a partially dehydrogenated mono-adduct as the key intermediate enabling the second cycloaddition. The resulting product represents a rare example of PAH diimide featuring an acceptor-donor-acceptor type structure. Detailed spectroscopic and theoretical studies, including transient absorption and two-dimensional electronic spectroscopy, revealed the emergence of a bright intramolecular charge transfer state that could be directly excited to demonstrate distinct photophysical dynamics. These findings provide deep mechanistic insights into the D–A reactivity of large PAHs and underscores the potential of such PAH imides for advanced optoelectronic and photonic applications.

Polycyclic aromatic hydrocarbons (PAHs) can offer high versatility in designing ad hoc molecular functions through engineering of their side groups, periphery topologies and extension of their π-conjugation. Amongst various types of PAH derivatives, PAH imides with acceptor-donor-acceptor (A-D-A) structures consisting of electron-rich aromatic cores and electron-withdrawing dicarboximide moieties have attracted particular attentions[1–4]. These molecular systems are expected to exhibit narrow optical energy gaps and pronounced photoinduced charge-transfer (CT) characters, giving rise to interesting photophysical features that range from long-lived charge-separated states to high polarizabilities, which are highly relevant in the fields of organic field-effect transistors, nonlinear optical materials, near-infrared dyes, and molecular wires[1,5–9]. While various types

of PAH imides have been reported, such as fused oligo(rylene diimide)[7,10–22] and tetrabenzoperipentacene tetraimide (Fig. 1a)[23], PAH imides with extended aromatic cores are still rare[24,25], and the possible presence of intramolecular charge transfer characters in such large A-D-A structures have received little attention.

Conventional synthetic methods for PAH derivatives typically involve constructing polyphenylenes through transition-metal-catalyzed cross-coupling reactions followed by cyclization reactions, such as cycloisomerization of *ortho*-ethynyl biaryls[26–29] and oxidative cyclodehydrogenation, namely Scholl reaction, of oligoarylenes[30]. While these methods require multiple synthetic steps, annulative π-extension (APEX) has recently attracted attention as an efficient approach to directly extend the π-systems

[1]Max Planck Institute for Polymer Research, Mainz, Germany. [2]Department of Chemistry, University of Oxford, Chemistry Research Laboratory, Oxford, UK. [3]State Key Laboratory of Bioinspired Interfacial Materials Science, Institute of Functional Nano & Soft Materials (FUNSOM) Soochow University, Suzhou, Jiangsu, PR China. [4]Physics Department, Politecnico di Milano, Milano, Italy. [5]División de Ciencias Básicas e Ingeniería, Departamento de Química, Universidad Autónoma Metropolitana-Iztapalapa, México, México. [6]Materials and Processes Simulation Lab, Centre of New Technologies, University of Warsaw, Warsaw, Poland. [7]Department of Chemistry, Johannes Gutenberg-University, Mainz, Germany. [8]Organic and Carbon Nanomaterials Unit, Okinawa Institute of Science and Technology Graduate University, Okinawa, Japan. [9]Present address: Department of Applied Science and Technology, Politecnico di Torino, Torino, Italy. ✉e-mail: fm@xanum.uam.mx; giuseppemaria.paterno@polimi.it; akimitsu.narita@oist.jp

## a) Representative PAH diimides[22,23]

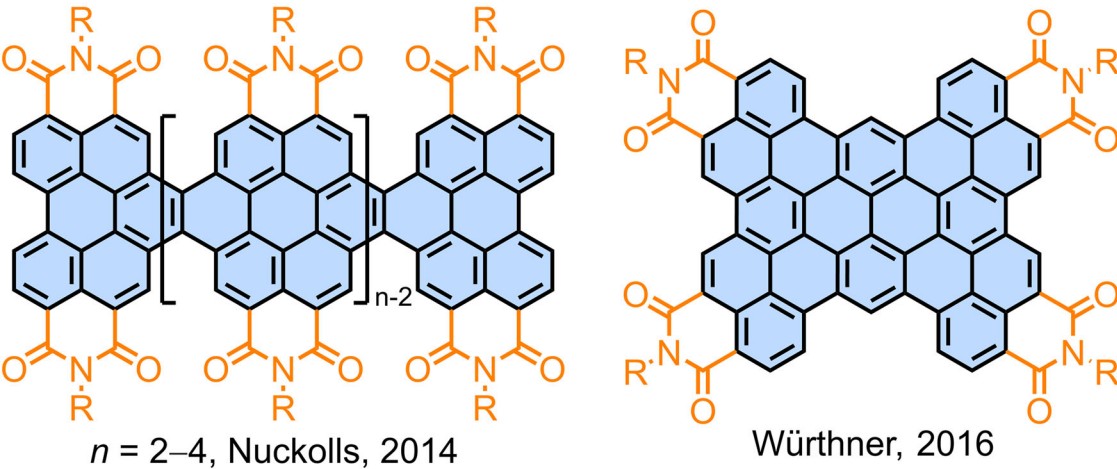

*n* = 2–4, Nuckolls, 2014

Würthner, 2016

## b) D-A addition of periacenes with 3 rows of fused benzene[37-42]

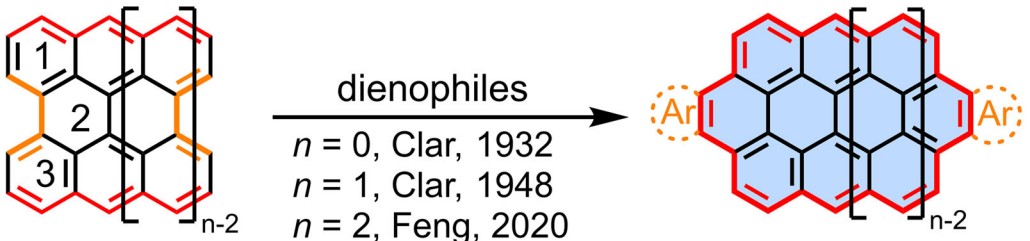

dienophiles

*n* = 0, Clar, 1932
*n* = 1, Clar, 1948
*n* = 2, Feng, 2020

## c) D-A addition of DBOV with 4 rows of fused benzene (this work)

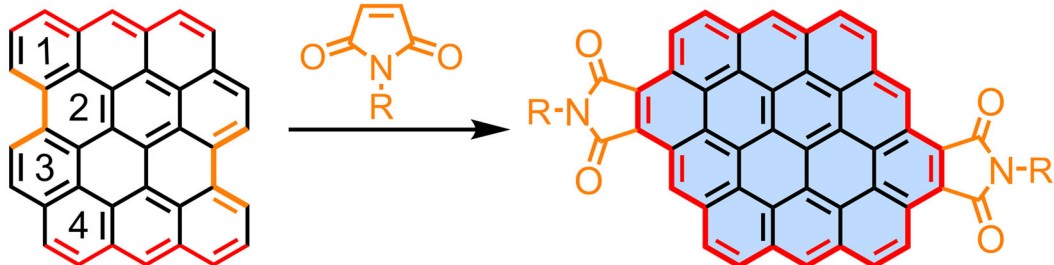

**Fig. 1 | Representative PAH diimides and synthesis methods. a** Fused perylene diimide oligomers[22] and tetrabenzoperipentacene tetraimide[23]; (**b**) D–A reaction of periacenes to circumacene derivatives[37–42]; (**c**) D–A reaction of DBOV with maleimide giving access to circumpyrene tetracarboxydiimide (this work).

without prior functionalization[31–33]. This defines a close similarity to the Diels–Alder (D–A) protocol[30,34–36]. Clar and Zander demonstrated that the bay region of perylene serves as the diene to react with maleic anhydride toward a tetrahydrobenzo[*ghi*]perylene derivative, followed by the oxidation to the fully aromatized structure[37–39]. More recent examples were reported by the groups of Scott[40], Wu[41] and Feng[42], achieving the D–A reaction of bisanthene and peritetracene as bisdienes, giving access to different circumacene derivatives (Fig. 1b) with unique optical and magnetic properties. However, the scope of suitable PAHs has been rather limited[43,44]. Despite the presence of numerous other PAHs featuring bay regions, including hexa-*peri*-hexabenzocoronene (HBC) and fused anthenes[30,45–48], extension of their π-systems through the D–A reaction has thus remained unsuccessful[49] or unexplored.

Dibenzo[*hi,st*]ovalene (DBOV, Fig. 1c), which has recently been synthesized by us, is a PAH featuring two bay regions and exhibits intriguing optical properties, such as stimulated emission (SE) and single-photon emission[50,51]. The π-extension at bay positions previously led to circumpyrene[52]. This synthesis required regioselective bromination[53,54], followed by palladium-catalyzed annulation with diphenylacetylenes or

Sonogashira coupling to introduce protected ethynyl groups, deprotection, and platinum-catalyzed cycloaromatization. In this work, we explored the D–A cycloaddition of DBOV, one of the largest PAH-based dienes thus far investigated, and accomplished its direct π-extension to circumpyrene tetracarboxydiimide. In contrast to periacenes, including perylene and bisanthene, DBOV features four rows of fused benzene rings (Fig. 1b, c) and belongs to an unusual class of PAH dienes in the D–A reaction. Interestingly, fully aromatized mono-adduct was isolated together with the diadduct even after reaction at 265 °C for 3 days, and the former was unreactive toward the D–A addition. This intriguing observation prompted us to carry out density functional theory (DFT) calculations to obtain detailed mechanistic insights into this two-fold D–A reaction. In contrast to our initial assumption that the tetrahydro derivative of the mono-adduct might be the key intermediate for the second cycloaddition, the results indicated that a partially dehydrogenated, dihydro derivative, was the plausible intermediate, accounting for the experimental observation. Circumpyrene tetracarboxydiimide is one of the largest PAH diimides thus far reported[22,23,42], and exhibited red-shifted absorption and emission with enhanced fluorescence, compared to the previous circumpyrene derivatives.

Remarkably, in-depth studies of its photophysical properties, involving ultrafast transient absorption spectroscopy and two-dimensional electronic spectroscopy, revealed an intriguing bright intramolecular CT state, highlighting the advantage of such imide-fused, A-D-A type structures.

## Results and discussion

### Synthesis and structural characterization

To investigate the D–A reaction of DBOV, a mesityl (Mes-) derivative DBOV-Mes **1** was selected (Fig. 2a)[50,55]. The reaction of **1** with 4-phenyl-1,2,4-triazole-3,5-dione (toluene, 110 °C, 6 h)[56], nitroethylene (135 °C, 20 h), and diethyl acetylenedicarboxylate (diphenyl ether, 260 °C, 20 h) did not provide any adduct, and the use of *p*-chloranil as oxidant, in an attempt to promote the dehydrogenation, led to decomposition of **1**. Nevertheless, formation of a mono-adduct was observed for the reaction with molten maleic anhydride at 260 °C, as indicated by mass spectrometry (see Fig. S7). Finally, the reaction of **1** with *N*-hexyl maleimide (**2a**) successfully gave circumpyrene tetracarboxydiimides **4a** as the di-adduct, along with mono-adduct **3a** in 18% and 16% yield, respectively, after heating at 265 °C for 3 days. We assume that the dehydrogenation of the D–A adducts was promoted by excess maleimide[57] and/or a trace of oxygen in the reaction system. Besides characterizations by MS (see Figs. S16 and S17), well-resolved ¹H NMR spectra of **3a** and **4a** could be obtained in $C_2D_2Cl_4$ at 413 K (Fig. 2b, c, see Supplementary Data 1 for the full spectra) and all the peaks could be fully assigned by 2D NMR techniques. *N*-(4-*tert*-butylbenzyl)maleimide (**2b**) showed similar reactivity to give mono-adduct **3b** and di-adduct **4b** in 9% and 4% yield, respectively. By contrast, the reaction with *N*-(4-*tert*-butylphenyl)maleimide (**2c**) provided only mono-adduct **3c** in 23% yield, presumably due to the conjugation between 4-*tert*-butylphenyl and maleimide moieties, making it a weaker dienophile.

### Investigation of the reaction mechanism

To understand the observed low D–A addition efficiency of DBOV, the activation energies of the D–A reactions with acetylene and *N*-phenylmaleimide were calculated in gas phase at the B3LYP/6-31 G(d,p) level of theory (Fig. S1)[40]. For comparison, the same calculations in gas phase and the same level of theory were also conducted for phenanthrene[40], the periacenes[40], and HBC with four K-regions, or so-called "tetrazigzag" HBC, as another recently reported PAH featuring two bay regions[50,58]. The activation energy of the reaction of DBOV with acetylene was slightly larger than that of perylene and smaller than phenanthrene that has never been reported to undergo a bay-region D–A addition, in agreement with the experimental observations. The activation energy of "tetrazigzag" HBC was calculated to be close to that of phenanthrene, in line with previous attempts for the D–A reaction that did not yield any adduct[49]. The activation energies of these PAHs for the reaction with *N*-phenylmaleimide show the same trend while their corresponding absolute values are relatively smaller than those for acetylene.

D–A reaction of mono-adduct **3a** with **2a** was also attempted after the isolation, but no di-adduct **4a** could be detected by MS after heating at 265 °C for 3 days. The lower reactivity of the second D–A addition was found for periacenes, including perylene and bisanthene[59]. The failed reaction of **3a** is due to a lowered HOMO level (from –4.49 eV for **1** to –4.85 eV for **3a** based on DFT calculation; *vide infra*). In order to quantitatively understand the observed lower reactivity, the potential energy surface of the D–A reaction of **1** and **2a** was calculated at different levels of theory (B3LYP/6-31G(d)//B3LYP/6-31G(d), M062X/6-31G(d)//B3LYP/6-31G(d) and ωB97XD/6-31G(d)//B3LYP/6-31G(d)) using diphenyl ether as solvent with the polarizable continuum model (PCM) (Section 3 of SI). We use three different functionals to observe the effect of the approximations on the exchange and correlation terms and on the dispersion forces in the study

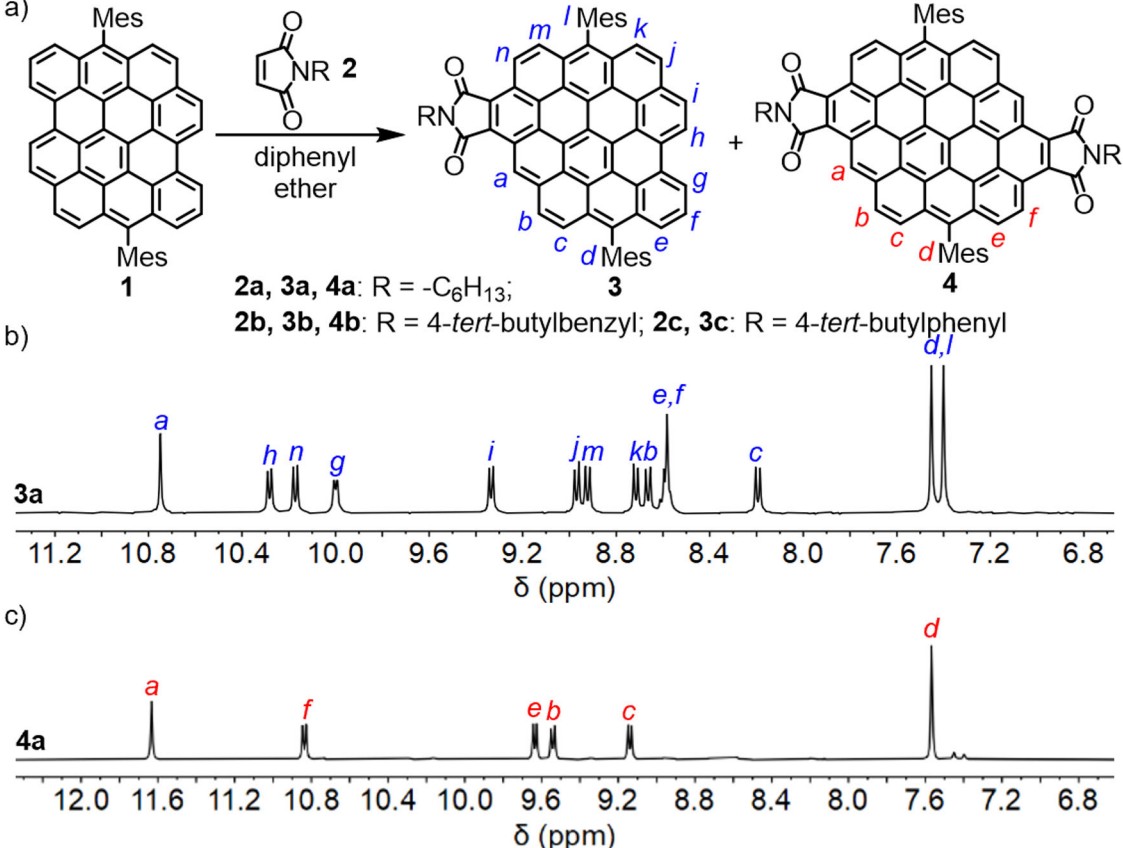

**Fig. 2 | Synthesis and characterization of circumpyrene tetracarboxydiimides. a** Synthesis of circumpyrene tetracarboxydiimides **4** through D–A reaction; (**b, c**) Aromatic regions of ¹H NMR spectra of **3a** and **4a** in $C_2D_2Cl_4$ (500 MHz, 413 K).

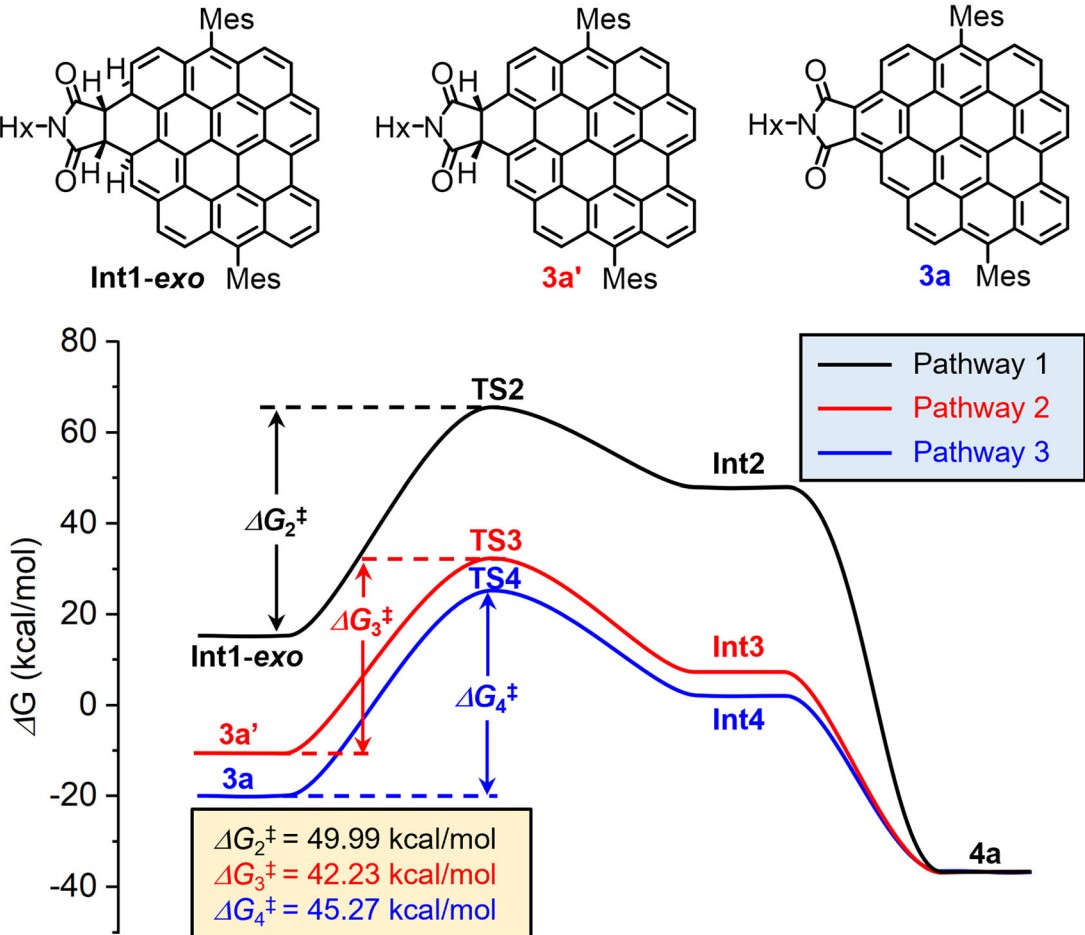

**Fig. 3 | The relative free energy profiles for the D–A addition reaction.** The relative free energy profiles for the D–A addition of **1** with *N*-hexyl maleimide (**2a**) in diphenyl ether via pathway 1 through **Int1-exo** and pathway 2/3 involving partially (**3a'**) and fully aromatized mono-adduct (**3a**), calculated at the B3LYP/6-31 G(d) //B3LYP/6-31 G(d) level of theory using the polarizable continuum model (PCM).

of our D–A reactions. The trend of the results is the same, so the three functionals are suitable for studying the D–A reaction. We discuss in the manuscript the results obtained with the functional B3LYP. For the first step to form **3a**, four stereoisomers with inner/outer and *exo-/endo*-structures were considered, among which the outer *exo*-addition gave the most stable transition state (**TS1**, Fig. S2) with a Gibbs free energy of activation ($\Delta G_1^{\ddagger}$) of 40.56 kcal/mol and reaction energy of −1.00 kcal/mol. The activation energy of *endo*-addition was 0.63 kcal/mol higher than that of the *exo*-addition with the reaction energy being 2.45 kcal/mol higher, so the *exo*-addition pathway was used for all the calculations. The Gibbs free energy of activation of the second D–A addition, namely **TS2** from intermediate **Int1-exo** to **Int2** (Fig. 3a, pathway 1) ($\Delta G_2^{\ddagger}$) was also calculated in *exo*-manner, which gave a higher value of 49.99 kcal/mol. By contrast, the Gibbs free energy of activation of the D–A reaction between **2a** and partially/fully dehydrogenated intermediates (Fig. 3, pathway 2/3, **3a'**/**3a**) exhibited a lower activation energy ($\Delta G_3^{\ddagger}$/$\Delta G_4^{\ddagger}$) of 42.23/45.27 kcal/mol, respectively. The fact that the activation energy of the D–A reaction through pathway 2 is smaller than through pathways 1 and 3 and larger than $\Delta G_1^{\ddagger}$ clearly agrees with the low yield of **4a** and the experimental observation that the D–A reaction of **3a** did not proceed even in the presence of 200 eq. of maleic imide. The calculation results also pointed toward **3a'** as the most plausible intermediate to form **4a**, which becomes less reactive after being fully oxidized to **3a**. Moreover, for the reaction of **1** and **2c**, one reaction intermediate with the same mass as partially aromatized one-fold D–A adduct **3c'** could be isolated, which was readily oxidized to **3c** upon addition of DDQ (Fig. S8). This partially dehydrogenated intermediate was not observed for the D–A addition between DBOV-Mes and **2a/b**, presumably due to the

stronger oxidizing ability of these two maleimides. Based on our mechanistic understanding of the whole reaction process, we speculate that any measures that contribute to accumulation of **3a'** and suppression of its aromatization to form **3a** would increase the yield of two-fold D–A adduct **4a**.

**Optoelectronic properties**

The optical properties of **3a/4a** and the effect of the dicarboximide groups were studied by UV-vis absorption and fluorescence spectroscopy (Fig. 4a, b). The lowest-energy absorption bands of the red solution of **3a/4a** are located at 610/628 nm, respectively, in agreement with TD-DFT calculations (Fig. S6). Their absorption maxima were located at 548/448 nm, with large molar extinction coefficient of $5.02 \times 10^4$/ $8.13 \times 10^4 \, M^{-1} \, cm^{-1}$, respectively. The increased extinction coefficient agreed with their higher oscillator strength compared with circumpyrene. The UV-vis absorption spectra showed no change after storing the toluene solution under ambient condition for 9 days, indicating their high stability.

Both **3a** and **4a** displayed bright red fluorescence in toluene solution with emission maxima at 631/637 nm, respectively (Fig. 4b). Compared with a previously reported circumpyrene derivative (**CP**, Fig. 4b) without imide groups, whose lowest-energy absorption and fluorescence peaks were located at 549/555 nm[52], the absorption and fluorescence spectra of **4a** were red-shifted, demonstrating its narrowed optical energy gap. The absolute fluorescence quantum yields ($\Phi$) of **3a** and **4a** were measured to be 0.49 and 0.61, respectively, five times higher than those of the analogs without imide groups[52]. The cyclic voltammograms of **3a/4a** measured in *o*-dichlorobenzene (Fig. 4c) revealed one reversible oxidation and two reduction

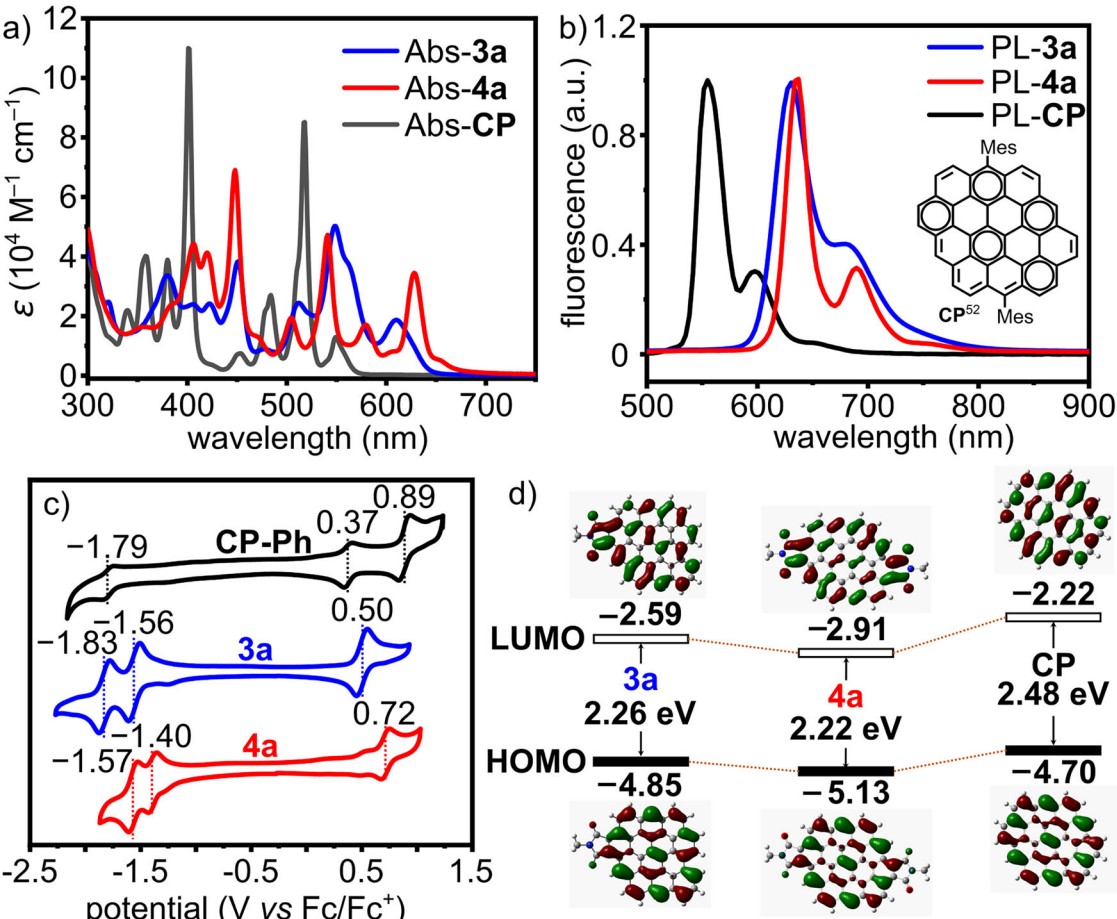

**Fig. 4 | Photophysical and electrochemical properties. a** UV-vis absorption and (**b**) fluorescence spectra of **3a**, **4a**, and **CP** in toluene ($c = 10^{-5}$ M, 25 °C, the spectroscopic data for **CP** were taken from ref. 52.); (**c**) Cyclic voltammograms of **3a** and **4a** measured in *o*-dichlorobenzene ($c = 0.1$ mM, 25 °C, scan rate = 50 mV/s), with 0.1 M tetra-*n*-butylammonium hexafluorophosphate as electrolyte in comparison with **CP-Ph**; (**d**) DFT-calculated frontier molecular orbitals and energy gaps of **3a**, **4a**, and **CP** at B3LYP/6-31 G(d,p) level of theory (Mes groups on molecular models were hidden for clarity). Source data underlying all the graphs are provided in Supplementary Data 2.

peaks, and indicate lower HOMO/LUMO energies[60] than those of the circumpyrene derivative with four phenyl groups (**CP-Ph**)[52], marking the electron-withdrawing effectiveness of the dicarboximide groups. The electrochemical energy gaps of **3a/4a** (1.90 and 1.74 eV, respectively) were also smaller than that of **CP-Ph** (2.16 eV), in line with DFT calculations (Fig. 4d)[52].

## Intramolecular charge transfer and transient absorption spectroscopy

To investigate the possible charge transfer character of **4a** with an A-D-A structure, its absorption and emission spectra were recorded in different solvents, which revealed an interesting solvent-polarity dependence (Fig. 5a and Fig. 5b). Specifically, we noted, at longer wavelengths with respect to the main long-wavelength transition, a broad and weak absorption band that is small yet noticeable in *n*-hexane (644 nm) and develops upon increase of solvent polarity. Furthermore, such a solvent-sensitive band redshifts from 644 nm in *n*-hexane to 652 nm in acetonitrile, which was consistent with TD-DFT calculation results (Fig. S6), although it is known to be a challenge to simulate the optical spectra of such PAHs[61]. Analogously, the photoluminescence of **4a** undergoes a clear broadening and redshift (42 nm) passing from *n*-hexane to acetonitrile. These broad, redshifted and featureless polarity-sensitive absorption and emission bands are typical signatures of a bright CT state[62,63]. Given the relatively high dilution of the solutions ($10^{-6}$ M) and relatively high solubility of **4a** in the selected solvents, we thus reckon

that this would be an intramolecular CT state rather than due to aggregation effects.

Additionally, we assessed the degree of the CT character using the Lambda parameter, which varies between 0 and 1, with small values corresponding to long-range (i.e., CT) excitations and large values to short range (local) excitations[64] (see the SI for details). For the $S_1$ transition simulated by the TD-DFT calculation, we obtained a value of 0.7 for **4a** in acetonitrile, and larger values of 0.77–0.78 in other solvents, suggesting a partial CT contribution in acetonitrile. The overlap between holes and electrons is around 0.75 for all cases, indicating the hybridization of LE and CT characters in all solvents, in line with the experimental results.

To obtain a deeper insight into the excited state dynamics and the possible intramolecular CT character, we carried out ultrafast broadband transient absorption (TA) spectroscopy experiments on **4a** in toluene solution. In general, the positive differential transmission ($\Delta T/T$) signals are linked to enhanced transmission of the probe pulse due to the pump-induced bleaching of the ground state absorption (photobleaching, PB), while the negative signals correspond to a reduced transmission of the probe owing to excited state absorption (ESA). The probe pulse can also stimulate emission from the excited states, showing a positive signal (SE). First, we excited in resonance with the main transition at 620 nm (Fig. 5c), observing a rich TA spectrum, with a broad ESA band extending in the visible and near-infrared regions and different PB signals (450, 505, 540, 575, and 620 nm). The increase of the PB and SE signals over time can be attributed to the picosecond rotational dynamics of the excited molecules, which is

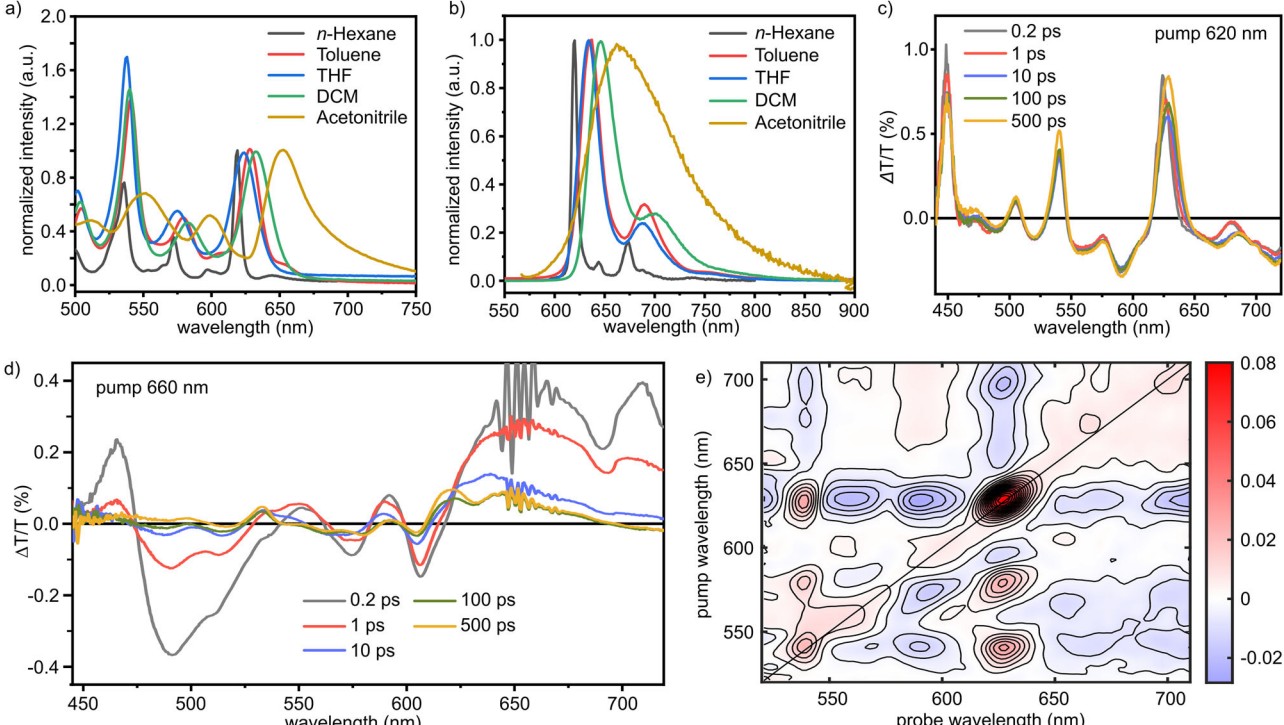

**Fig. 5 | Solvent-dependent steady state photophysical properties and transient absorption of 4a. a** UV-vis absorption and (**b**) photoluminescence of **4a** dissolved in different solvents at a concentration of $7.5 \times 10^{-6}$ mol/L. The photoluminescence spectra were taken by exciting at the maximum absorption wavelength. THF tetrahydrofuran, DCM dichloromethane; (**c**) Transient absorption spectra of **4a** upon excitation at 620 nm (toluene, 0.1 mg/mL) at different delay times between pump and probe and (**d**) upon excitation at 660 nm, with the noisy region at probe wavelengths around 660 nm, which might resemble vibronic-like features, resulting from pump scattering; (**e**) Two-dimensional electronic spectroscopy map taken at a delay between pump and probe of 1 ps. Source data underlying all the graphs are provided in Supplementary Data 3.

captured due to the cross-polarized pump-probe configuration adopted in our experiment[65,66]. Namely, the linearly polarized pump pulse in the setup preferentially excites molecules with parallelly or nearly parallelly aligned transition dipole moments. The probe beam, which is polarized perpendicular to the pump, initially probes molecules whose emission or absorption transition moments are orthogonal to those that were preferentially excited. As the excited molecules in solution undergo Brownian rotational motion in the picosecond scale, the initially photoselected population randomizes and the anisotropy decays, so the PB and SE signal can be detected and increase in intensity. Furthermore, we observed a redshift of the SE upon increasing of the time-delay (Fig. S13), which can be attributed to the conformational rearrangement happening in the lowest excited state to attain a planar and fully conjugated geometry[54].

To selectively probe the state with probable CT character, we excited **4a** at 660 nm in toluene, i.e., on the red shoulder of the lowest absorption band (Fig. 5d). In this case the transient response is markedly different from that observed at 620 nm. Broad positive features appear at 551, 592 and 660 nm, accompanied by a positive band around 710 nm and a pronounced negative band near 490 nm. The bands at 551, 592 and 660 nm coincide with the low-energy absorption features of **4a** in polar solvents and are therefore assigned to ground-state bleach of the bright CT-like transition, whereas the 710 nm signal overlaps with the broad emission band in dichloromethane (DCM) and acetonitrile and is attributed to SE from the CT-like state. The negative feature at 490 nm is assigned to ESA from this state. To quantify the underlying dynamics, we performed a three-component global analysis of the 660 nm data (Fig. S15). The spectra are well reproduced by an instrument-response-limited component with $\tau_1 = 0.20 \pm 0.05$ ps, a second component with $\tau_2 = 4.7 \pm 1.5$ ps, and a long-lived component with $\tau_3 > 500$ ps. The decay-associated spectrum (DAS) of the sub-ps component closely resembles the earliest measured spectrum and accounts for coherent artefacts and ultrafast vibrational/solvent relaxation, for which we do not

attempt a detailed assignment. The $\tau_2$ component, in contrast, displays pronounced spectral reshaping with opposite sign relative to the long-lived DAS in several regions, which is characteristic of population transfer between two distinct excited configurations. The $\tau_3$ DAS matches the spectrum measured at 500 ps after 660 nm excitation and thus represents the relaxed CT-rich excited state. Consecutive and parallel kinetic models yield essentially identical lifetimes and DAS, confirming the robustness of the extracted time constants. The absence of any comparable picosecond evolution in the 620 nm data, where the spectra remain virtually unchanged up to 500 ps, indicates that this CT-rich state is not efficiently populated upon excitation of the LE state, but is instead selectively accessed upon red-edge excitation at 660 nm.

Finally, we investigated the excited state dynamics of **4a** in toluene employing two-dimensional electronic spectroscopy[67] (2DES, Fig. 5e). This technique enables us to resolve the pump wavelength dependence of the TA signal, allowing observation of the system's response at every frequency within the pump pulse bandwidth. In 2DES, the diagonal peaks correspond to the probe at the same energy as the pump, reflecting information on linear absorption and linewidth broadening mechanisms. Conversely, off-diagonal peaks, or cross-peaks, indicate coupling between two electronic states of the system. Figure 5e presents a 2DES map at a delay of 1 ps, utilizing identical pulses for both pump and probe, with excitation/detection wavelengths covering a range from 520 to 710 nm, thus enabling simultaneous investigation of the local and the CT excitation. Here we exploit the 2DES capability to resolve the pump-wavelength dependence of the transient signal with high resolution. A horizontal cut of the map effectively allows us to visualize the system's response to every excitation frequency within the pulse bandwidth, acting as a series of narrowband pump-probe experiments in a single measurement (i.e., at 630 nm, see Fig. S14a). In particular, for excitation at 630 nm one observes a diagonal peak corresponding to the LE transition, elongated along the diagonal direction due to

inhomogeneous broadening. Note that the persistence of such broadening is typical of nanographenes, suggesting that different microscopic environments cause shifts in the otherwise narrow absorption lines of individual chromophores (Fig. S14c, d)[68,69]. Crucially, scanning the excitation axis to lower energies (>640 nm) reveals a starkly different photophysical regime. Here, a broad, positive, and red-shifted off-diagonal signal appears (centered at probe ~660 nm). The spectral position and width of this off-diagonal feature in the 2DES map coincide with the long-lived DAS obtained from the global analysis of the 660 nm pump–probe data (Fig. S15), providing independent confirmation that the $\tau_3$ component corresponds to the bright CT-like state. The clear separation of these regimes in the 2D map provides direct evidence that the CT state is not a dark manifold populated only by relaxation from $S_{LE}$, but a bright electronic state with its own distinct transition dipole. This underscores the power of 2DES to disentangle spectrally overlapping excited states and confirms that imide functionalization indeed generates a genuinely bright, optically addressable intramolecular charge-transfer state.

It is interesting to note that in DBOV derivatives we observed the development of dark CT states in solid films and concentrated solutions, which in those cases were assigned to ultrafast electron transfer occurring in the intermolecular aggregates, leading to dramatic quenching of the emission and optical gain[51,70,71]. Conversely, **4a** affords a bright intramolecular CT state that likely originates from hybridization of locally excited and CT states[72–75]

This leads to considerable extension of its optical absorption and emission into the deep-red region as well as enhanced fluorescence quantum yields. These results have important technological implications, since direct excitations of bright CT states is extremely relevant for boosting the efficiency of photovoltaic diodes, as they ultimately mediate the conversion of light energy into charges, and the occurrence of intramolecular singlet fission pathways[62,76–78].

## Conclusion

In conclusion, we have investigated the synthesis of circumpyrene tetracarboxydiimides through two-fold D–A cycloaddition at the bay regions of DBOV with maleimides. By using DFT calculations, we found that the partially dehydrogenated D-A mono-adduct **3a'** showed lower activation energy toward the D-A cycloaddition than the fully aromatized mono-adduct **3a**, but still higher than that for the first D-A addition of DBOV-Mes, rationalizing the experimentally observed low yield of two-fold D-A adducts. This result suggests that full aromatization of the mono-adduct intermediate should be suppressed to obtain more of the di-adduct. We observe that the imide groups red-shifted and intensified the fluorescence of the parent circumpyrene. Moreover, TA and 2DES measurements revealed the presence of a bright intramolecular charge transfer state, which is highly relevant for the operation of optoelectronic devices. Furthermore, the imide groups could be potentially functionalized for different purposes, e.g., further modulation of the photophysical properties and water solubilization for bioimaging.

## Methods

All experimental details and methods are given in the Supplementary Information.

## Data availability

Experimental and theoretical details, including synthetic protocols, MALDI-TOF MS spectra, Supplementary figures and tables of calculation results, additional UV-vis absorption, fluorescence, and transient absorption spectra, and Cartesian coordinates of optimized structures are available in the Supporting Information file attached to this paper. NMR spectra are provided as Supplementary Data 1. The source data for the graphs in Fig. 4 and Fig. 5 are provided as Supplementary Data 2 and 3, respectively. Other source data are available at Zenodo (https://doi.org/10.5281/zenodo.18298550) or from the corresponding authors on reasonable request.

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

## Acknowledgements
This work was financially supported by the Max Planck Society, the ANR-DFG NLE Grant GRANAO by DFG 431450789, and the European Union's Horizon 2020 Research and Innovation program under grant agreement no. 101017821 (LIGHT-CAP). Q.C. is thankful for the support from Suzhou Key Laboratory of Functional Nano & Soft Materials, Collaborative Innovation Center of Suzhou Nano Science & Technology, the 111 Project, Suzhou Key Laboratory of Surface and Interface Intelligent Matter (No. SZS2022011), National Natural Science Foundation of China (22405187), Gusu Innovation and Entrepreneurship Leading Talent Program (ZXL2024391), and Natural Science Foundation of Jiangsu Province (BK20240759). K.M. acknowledges a fellowship from Gutenberg Research College, Johannes Gutenberg University Mainz. A.N. appreciates the financial support from the Okinawa Institute of Science and Technology Graduate University. S.O. thanks the Polish National Science Centre for funding (grant no. UMO-2023/50/E/ST4/00197). G.M.P. receives funding from the European Union (ERC, EOS, 101115925). G.C. and G.M.P. acknowledge financial support by the European Union's Next Generation EU Program with the IPHOQS Infrastructure [IR0000016, ID D2B8D520, CUP B53C22001750006] "Integrated Infrastructure Initiative in Photonic and Quantum Sciences".

## Author contributions
Q.C. carried out the synthesis and basic characterizations under the supervision by K.M. and A.N. F.M. conducted the theoretical investigations of the mechanism of the Diels-Alder reaction. M.G. measured the transient absorption and two-dimensional electronic spectroscopy under the supervision by G.C., F.S., and G.M. P.H.K. did the global analysis of the TA data. C.J. synthesized the dialdehyde intermediate for DBOV-Mes under the supervision by K.M. and A.N. Q.C., and S.O. performed the theoretical studies of the optical properties. Q.C., F.M., G.M.P. and A.N. prepared the manuscript with contributions by other authors. All authors discussed the results and commented on the manuscript.

## Funding

## Competing interests
The authors declare that Q.C, K.M., and A.N. are listed as inventors on two patents 1) PCT/EP2019/076496, WO2020070085 and 2) PCT/EP2019/076497, WO2020070086 related to the work presented in this manuscript. All other authors have nothing to disclose.
