## [Transparent Peer Review file · Communications Chemistry]

Diels-Alder reaction affords circumpyrene tetracarboxydiimide with excited state intramolecular charge transfer character

Corresponding Author: Professor Akimitsu Narita

Version 0:

Reviewer comments:

Reviewer #1

(Remarks to the Author)

The manuscript "Diels-Alder Reaction Towards Circumpyrene Tetracarboxydiimide with Excited State Intramolecular Charge Transfer Character" submitted by Narita et al. offers a very interesting insight into the chemistry of polycyclic aromatic hydrocarbons. The authors have performed double Diels-Alder cycloadditions of N-hexyl maleimide to dibenzo[*hi,st*]ovalene and characterized the reaction products very thoroughly. Interestingly, the authors showed that an intermediate compound - didehydrogenated mono-adduct 3a' - was likely responsible for the second Diels-Alder reaction and not fully dehydrogenated compound 3a. This assumption is strongly supported by extensive high-level DFT calculations. Optical spectroscopies, transient absorption spectroscopies, and cyclic voltammetry allowed for the determination of optoelectronic properties of circumpyrene tetracarboxydiimides. These materials are interesting candidates for various devices and applications for NIR dyes. This manuscript will find large attention from researchers in the fields of materials science, nanographenes, and polycyclic aromatic hydrocarbons in general. I recommend publication in Communications Chemistry without any changes.

Reviewer #2

(Remarks to the Author)

Here the authors have investigated the synthesis of circumpyrene tetracarboxydiimides by D-A cycloaddition of dibenzo[*hi,st*]ovalene, one of the largest PAHs functioning as bisdiene, with maleimides. Using Density functional theory (DFT) calculations, they have proposed a partially dehydrogenated mono-adduct to be the key intermediate enabling the second cycloaddition as the fully aromatized mono-adduct is inert toward further D-A reaction. The introduction of imide functionalities led to enhanced and red-shifted fluorescence compared to the parent circumpyrene. Utilizing transient absorption and two-dimensional electronic spectroscopy, the authors have claimed to observe the formation of a bright intramolecular charge-transfer state, which gave rise to long-wavelength absorption and stimulated emission in the near-infrared region that can be promising for advanced optoelectronic and photonic applications.

1. Synthesizing this large PAH D-A system is always challenging. All these reactions were executed at high temperatures while the reaction timescales are in few days without a significant reaction yield. As the authors have followed the reaction mechanism using DFT, can they propose plausible modifications that can enhance the yield further in order to give a broader perspective?
2. In the synthesis of circumpyrene tetracarboxydiimides 4a and 4b, what is mechanism of dehydrogenation after Diels Alder addition to regain aromaticity? Is there any role of the solvent?
3. The absolute fluorescence quantum yields of 3a and 4a were measured to be around five times higher than those of the analogues without imide groups. Generally, CT states give rise to lower QY. What is so special in these systems to enhance the QY by 5 times?
4. Can authors explain why vibronic-like features are observed in GSB of 4a when 660 nm pump pulse used (Figure 5d)?

5. "The increase of the PB and SE signals over time can be attributed to the picosecond rotational dynamics, which is captured due to the cross-polarized pump-probe configuration adopted in our experiment", can the authors elaborate on this statement?

6. How exactly have the authors assigned the CT state in TA spectra? What is the rationale behind that? An unequivocal assignment is required.

7. The authors need to do a deeper analysis of the TA and 2-D data. The assignment and explanations are not satisfactory as they do not explain the dynamics and spectra clearly enough.

I see that the authors have done a lot of experiments but all of them do not lead to a coherent story. I suggest major revisions and resubmission.

Reviewer #3

(Remarks to the Author)

The manuscript by Chen et al. describes the reaction of an extended polycyclic aromatic hydrocarbon (PAH) with maleimide derivatives through a Diels-Alder reaction localized to the two bay areas of the ovalene skeleton. This allows direct conversion of the PAH to the corresponding tetracarboxydiimide in one step without passing through the anhydride as is sometimes the case. During the synthesis, the authors note some peculiar behavior in that the mono-adduct, formed in equal yield, is unreactive towards further conversion to the diadduct. This is rationalized through computational modelling of the reaction to determine the activation energies for transformation of the plausible intermediates. Further photophysical characterization of the mono- and di-adducts provides some fundamental properties that could be of use to the community. Not surprisingly, given the expertise of the authors in this area, the research is conducted with care and the results discussed in the frame of the broader context of PAH chemistry. The real question for this reviewer is the general direction of this research: the topics covered in the introduction are noble (OFET, NLO, NIR), but it is hard to see how they can be impacted with compounds prepared in < 20% isolated yield from already complex starting materials. The small HOMO-LUMO energy gap is quite common for extended PAH, and the observation of ICT not unexpected upon introduction of acceptor groups. Perhaps more useful to the community would be a comparison of the calculated vs. experimental spectroscopic properties (to assist in future computational design) and the singlet-triplet energy gap (for TADF applications). The latter is more easily calculated than measured, but perhaps the observation of delayed fluorescence could show this. The more interesting aspects of the transient spectroscopy will be published separately, which sort of weakens the impact of this work. It can, however, be published in its present form and without any modifications if it is of sufficient interest for the journal.

Version 1:

Reviewer comments:

Reviewer #2

(Remarks to the Author)

WE thank the authors for giving a detailed point-by-point response and certainly improving the manuscript for scientific clarity. This reviewer is ok with the revisions although few minor improvements are suggested as below:

1. For the REPLY to our question number 2, the authors write the text:

"As to the dehydrogenation mechanism, we believe that excess amount of maleimide play role as an oxidant in the dehydrogenation process. Indeed, in a previously reported work (Org. Lett. 2003, 5, 2833), maleimide derivatives were also found to function as acceptor for hydrogen in D-A addition reaction."

If this were true will the dehydrogenation reaction not work when maleimide is a LIMITING reagent without any O2?

2. For the REPLY to our question number 7, the authors write the text:

"In this manuscript, our goal is to report and discuss the key spectral features and dynamical trends related to the presence of a bright CT state (as detailed in the previous answer). We feel that a deeper quantitative analysis, particularly of the ultrafast kinetic components and crosspeak evolution, will be the subject of a follow-up study aimed at a specialized readership in optical spectroscopy. Nonetheless, we believe that the additional clarifications we have added to address the points 4–6 from this reviewer have made the assignments and explanations clear in the revised manuscript."

This reviewer is not fully satisfied by this answer since it seems that 2D data was just shown to artificially enhance the impact of the work. I suggest the authors to remove this part in all seriousness. Already CT evidences have been provided by other techniques.

Reviewer #3

(Remarks to the Author)

The authors have commented on my remarks, but did not bring substantial modifications of the manuscript. While I still believe that, without the additional spectroscopic data, this work will have little impact outside the field, it can be published in its present format.

Response to Reviewers' Comments

Reviewers' comments:

Reviewer #1 (Remarks to the Author):

The manuscript "Diels-Alder Reaction Towards Circumpylene Tetracarboxydiimide with Excited State Intramolecular Charge Transfer Character" submitted by Narita et al. offers a very interesting insight into the chemistry of polycyclic aromatic hydrocarbons. The authors have performed double Diels-Alder cycloadditions of *N*-hexyl maleimide to dibenzoovalene and characterized the reaction products very thoroughly. Interestingly, the authors showed that an intermediate compound - didehydrogenated mono-adduct **3a'** - was likely responsible for the second Diels-Alder reaction and not fully dehydrogenated compound **3a**. This assumption is strongly supported by extensive high-level DFT calculations. Optical spectroscopies, transient absorption spectroscopies, and cyclic voltammetry allowed for the determination of optoelectronic properties of circumpylene tetracarboxydiimides. These materials are interesting candidates for various devices and applications for NIR dyes. This manuscript will find large attention from researchers in the fields of materials science, nanographenes, and polycyclic aromatic hydrocarbons in general. I recommend publication in *Communications Chemistry* without any changes.

Reply: We are grateful to the reviewer for his/her recognition of the importance of our work and recommendation for its publication in *Communications Chemistry*.

Reviewer #2 (Remarks to the Author):

Here the authors have investigated the synthesis of circumpylene tetracarboxydiimides by D-A cycloaddition of dibenzo[*hi,st*]ovalene, one of the largest PAHs functioning as bisdiene, with maleimides. Using Density functional theory (DFT) calculations, they have proposed a partially dehydrogenated mono-adduct to be the key intermediate enabling the second cycloaddition as the fully aromatized mono-adduct is inert toward further D-A reaction. The introduction of imide functionalities led to enhanced and red-shifted fluorescence compared to the parent circumpylene. Utilizing transient absorption and two-dimensional electronic spectroscopy, the authors have claimed to observe the formation of a bright intramolecular charge-transfer state, which gave rise to long-wavelength absorption and stimulated emission in the near-infrared region that can be promising for advanced optoelectronic and photonic applications.

Reply: We sincerely thank the reviewer for his/her careful assessment of our work.

1. Synthesizing this large PAH D-A system is always challenging. All these reactions were executed at high temperatures while the reaction timescales are in few days without a significant reaction yield. As the authors have followed the reaction mechanism using DFT, can they propose plausible modifications that can enhance the yield further in order to give a broader

perspective?

Reply: We appreciate the reviewer for raising this point. According to our DFT calculation results shown in Figure 3 in the manuscript, the activation energy of the Diels-Alder (D-A) cycloaddition of partially dehydrogenated intermediate **3a'** with maleic imide is smaller than that of the D-A adduct **Int1-exo** and fully aromatized mono-adduct **3a**. Consequently, if **Int1-exo** could be partially dehydrogenated to **3a'** without further dehydrogenation to **3a**, then the yield of bis-adduct **4a** could be enhanced in principle. We hypothesize that adding extra oxidant or dehydrogenation catalysts with appropriate oxidation potential that could selectively dehydrogenate **Int1-exo** to **3a'** might help to increase the yield of the circumpyrone tetracarboxydiimide (Scheme R1, strategy 1). The other option is to change the type of dienophiles, for example, to dichloromaleimide or dideuterated maleimides. After D-A addition on one side followed by partial dehydrogenation, the tendency of the resulting intermediates (**3a''**) to undergo further aromatization will be hampered and high reactivity towards the second D-A addition will be sustained (Scheme R1, strategy 2). Both of these two measures are expected to enhance the yield of D-A addition.

Scheme R1. Strategies to enhance the yield of D-A addition reaction.

Action: To discuss the DFT calculation results in more depth and make full use of our understanding on the reaction mechanism to improve our synthetic work, we made the following revision in the manuscript. On page 7, we added “Based on our mechanistic understanding of the whole reaction process, we speculate that any measures that contribute to accumulation of **3a'** and suppression of its aromatization to form **3a** would increase the yield of two-fold D-A adduct **4a**.” to the end of the mechanism investigation part.

2. In the synthesis of circumpyrone tetracarboxydiimides **4a** and **4b**, what is mechanism of dehydrogenation after Diels Alder addition to regain aromaticity? Is there any role of the solvent?

Reply: We sincerely appreciate the reviewer for this comment and question. As discussed above, we realized that partial dehydrogenation of the mono-side D-A adduct to **3a'** and preventing its further aromatization are essential for the successful synthesis of circumpyrone tetracarboxydiimides. Existence of oxygen in the reaction mixture might result in its full aromatization to **3a** so that further reaction will be unfavorable in energy. So, when we

performed the D-A addition, the solvent was carefully degassed by three times freeze-pump-thaw cycles and the reaction was performed under an inert atmosphere (argon) to exclude the effect of residual oxygen. As to the dehydrogenation mechanism, we believe that excess amount of maleimide play role as an oxidant in the dehydrogenation process. Indeed, in a previously reported work (*Org. Lett.* **2003**, *5*, 2833), maleimide derivatives were also found to function as acceptor for hydrogen in D-A addition reaction.

Action: In our previous manuscript we briefly explained the possible aromatization route of **3a'** to **3a** but not mentioned the bis-adduct. In the revised version, we have made it clearer by rephrasing the sentence to “We assume that the dehydrogenation of the D-A adducts was promoted by excess maleimide⁵⁷ and/or a trace of oxygen in the reaction system.” and moving it to 2.1. Synthesis and Structural Characterization on page 5.

3. The absolute fluorescence quantum yields of **3a** and **4a** were measured to be around five times higher than those of the analogues without imide groups. Generally, CT states give rise to lower QY. What is so special in these systems to enhance the QY by 5 times?

Reply: We thank the referee for this valuable question. Although charge-transfer (CT) states are often associated with low fluorescence quantum yields due to small oscillator strengths and enhanced nonradiative decay, bright CT states, where donor and acceptor orbitals retain sufficient overlap (thus exhibiting relatively high oscillator strength), can display high photoluminescence quantum yields (PLQY). In such systems, radiative rates remain significant while nonradiative decay channels are suppressed by structural rigidity, favorable excited-state geometry, or hybridization with locally excited (LE) character. In the case of **3a** and **4a**, the charge transfer character can be mixed with a strongly localized excitation, leading to their higher PLQYs. This is related to some fluorophores used for OLEDs (*Chem. Sci.*, **2021**, *12*, 5171) and to established lipid membrane targeting fluorescence probes, such as Laurdan (*J. Phys. Chem. B*, **2021**, *125*, 10748).

Action: To additionally explain this interesting photophysical phenomenon, we have revised the last paragraph before the conclusion part in page 12 as: “Conversely, **4a** affords a bright intramolecular CT state that likely originates from hybridization of locally excited and CT states^{72,73}. This leads to considerable extension of its optical absorption and emission into the deep-red region as well as enhanced fluorescence quantum yields.”

4. Can authors explain why vibronic-like features are observed in GSB of **4a** when 660 nm pump pulse used (Figure 5d)?

Reply: We thank the referee for this question. Such features are due to interferences, resulting from scattering of the strong pump beam inside the spectrometer, producing characteristic patterns that are instrumental artifacts rather than sample-related signals

Action: In the revised manuscript, on page 10 ((Fig. 5d caption)), we explained this feature by adding “(with the noisy region at probe wavelengths around 660 nm, which might resemble vibronic-like features, resulting from pump scattering)”.

5. “The increase of the PB and SE signals over time can be attributed to the picosecond rotational dynamics, which is captured due to the cross-polarized pump-probe configuration adopted in our experiment”, can the authors elaborate on this statement?

Reply: We thank the reviewer for requesting clarification of this important technical point. We elaborate as follows: In our transient absorption experiment, we employed a cross-polarized configuration where the pump and probe beams have perpendicular linear polarizations. This geometry introduces sensitivity to orientational dynamics through the phenomenon of photoselective excitation. The linearly polarized pump pulse preferentially excites molecules whose transition dipole moments are aligned parallel (or nearly parallel) to the pump polarization. This creates an anisotropic distribution of excited molecules immediately after excitation.

Since our probe beam is polarized perpendicular to the pump, it initially probes molecules whose emission/absorption transition moments are orthogonal to those that were preferentially excited. This results in an initially reduced TA signal compared to what would be observed in a parallel or magic-angle configuration. On the picosecond timescale, molecules undergo Brownian rotational motion in solution. As the initially photoselected population randomizes through rotational diffusion, the anisotropy decays. This means that excited molecules, which were initially aligned with the pump polarization, rotate and their transition moments become more randomly oriented [Refs: *PNAS*, **2022**, 119, e2117398119; *Photosynth. Res.*, **2009**, 101, 105–118].

Action: To make it clearer for the readers to understand this statement, in the revised manuscript, on page 10, we have cited two references (*PNAS*, **2022**, 119, e2117398119; *Photosynth. Res.*, **2009**, 101, 105–118) introducing the mechanism of this technique after this claim. On the other hand, we also expanded our previous description to “The increase of the PB and SE signals over time can be attributed to the picosecond rotational dynamics of the excited molecules, which is captured due to the cross-polarized pump-probe configuration adopted in our experiment.^{65, 66} Namely, the linearly polarized pump pulse in the setup preferentially excites molecules with parallelly or nearly parallelly aligned transition dipole moments. The probe beam, which is polarized perpendicular to the pump, initially probes molecules whose emission or absorption transition moments are orthogonal to those that were preferentially excited. As the excited molecules in solution undergo Brownian rotational motion in the picosecond scale, the initially photoselected population randomizes and the anisotropy decays, so the PB and SE signal can be detected and increase in intensity.”

6. How exactly have the authors assigned the CT state in TA spectra? What is the rationale behind that? An unequivocal assignment is required.

Reply: We thank the reviewers for this important question. We provide below a comprehensive rationale for our CT state assignment based on multiple independent lines of evidence:

(1) Solvatochromic Behavior (Steady-State Evidence)

Absorption: The longest-wavelength absorption band exhibits clear positive solvatochromism, red-shifting from 644 nm in *n*-hexane to 652 nm in acetonitrile. This solvent-polarity dependence is a hallmark signature of CT state formation.

Emission: The photoluminescence shows even more pronounced solvatochromic behavior with a 42 nm red-shift from *n*-hexane to acetonitrile, accompanied by spectral broadening and loss of vibronic structure. These are classic signatures of excited-state charge redistribution.

(2) Wavelength-Dependent Transient Absorption (Dynamic Evidence)

This is the most compelling evidence for distinct electronic states:

Local Excitation (620 nm): When pumping the main absorption band, we observe canonical TA features including multiple photobleaching signals at 450, 505, 540, 575, and 620 nm corresponding to ground-state bleaching (GSB), broad excited-state absorption (ESA) in the visible-NIR region, and stimulated emission (SE).

CT Excitation (660 nm): When pumping specifically at 660 nm (the weak shoulder peak), we observe a completely different TA spectrum:

1. Broad positive bands centred at 551, 592, and 660 nm (GSB of the CT absorption), which apparently correspond to absorption bands in acetonitrile
2. An additional positive signal at 710 nm (SE from the CT state), which coincide with the vibronic emission band in dichloromethane and overlap with the broad emission band in acetonitrile
3. A strong ESA band at 490 nm (characteristic of the CT state ESA)

The fact that direct excitation at different wavelengths produces fundamentally different TA signatures unambiguously demonstrates the presence of two distinct electronic states with different transition dipole moments and optical properties. Moreover, the similarity of the GSB and features after the excitation at 660 nm to the absorption and emission spectra in acetonitrile corroborate the assignment to the CT state.

The 2DES measurements provide the most unambiguous spectroscopic fingerprint:

Excitation-Wavelength Selectivity: Horizontal slices of the 2DES map at different excitation

wavelengths show:

1. Excitation at <640 nm produces the characteristic LE response with diagonal peaks
2. Excitation at >640 nm produces a drastically different response with a broad, red-shifted positive peak that cannot be observed at shorter excitation wavelengths

Direct CT State Identification: The appearance of this unique positive, broad signal only upon excitation at >640 nm provides direct confirmation that we are selectively exciting a distinct electronic state with its own characteristic optical response

While DBOV derivatives exhibit dark intermolecular CT states in aggregates that quench emission, compound **4a** displays a bright intramolecular CT state with non-zero oscillator strength. This distinction is critical: the CT state can be directly excited optically and produces SE, confirmed by both TA and 2DES measurements.

In addition, time-dependent density functional theory (TD-DFT) calculations also confirm the CT character of the S₁ transition, which involves excitation from the HOMO to the LUMO. From the calculations we do not see the broad band as experimentally observed in acetonitrile, and it is known to be difficult to precisely predict the optical properties of such PAHs, for which different theoretical methods can give contradicting results [*J. Chem. Phys.* **2019**, *150*, 124302]. However, we are able to reproduce the redshift in adsorption while changing the solvent polarity, with a maximum shift of 6 nm, which is in line with the experimentally observed 8 nm shift. As visible in Figure R2, a different localization is present for these two molecular orbitals with some overlaps; while the HOMO of **4a** is mainly localized on the core, the LUMO is significantly more localized on the outer parts including the imide moieties. This different spatial localization can give rise to the bright CT state for this transition.

Moreover, we have additionally assessed the degree of CT or local excitation using the Lambda parameter [*J. Chem. Phys.*, **2008**, *128*, 044118]. In this analysis, Lambda value varies between 0 and 1, with small values referring to long-range (i.e. CT) excitations and large values to short range (local) excitations. For this particular transition, we compute a value of 0.7 for **4a** in acetonitrile, and larger values of 0.77–0.78 when in other solvent, suggesting that indeed in acetonitrile there is a partial CT contribution to the transition. In addition, the overlap between holes and electrons is around 0.75 for all cases, meaning that there is a degree of CT in all solvents, also in agreement with the experimental results.

Figure R2. HOMO and LUMO of compound **4a** in acetonitrile.

Action: In order to better clarify the reasonable assignment of CT states in the TA spectra, we added “Here, we see a completely different spectroscopic response, with the appearance of broad positive bands centered at 551, 592, and 660 nm, which coincide with the absorption bands of **4a** in acetonitrile and thus assigned to GSB of the bright CT state. An additional positive signal is visible at 710 nm, which significantly overlaps with the vibronic band in the fluorescence spectrum of **4a** in dichloromethane (DCM) as well as its broad emission band in acetonitrile, and is thus assignable to SE of the CT state. Additionally, a strong ESA band is observed at 490 nm, which can also be characteristic of the bright CT state.” in the manuscript on page 11 and “To obtain a deeper insight into the excited state dynamics and the possible intramolecular CT character,” in the beginning of the TA part on page 10 to improve the logical connection there, regarding the investigation of the possible CT character.

For the TD-DFT calculation results, we have added “Furthermore, such a solvent-sensitive band redshifts from 644 nm in *n*-hexane to 652 nm in acetonitrile, which was consistent with TD-DFT calculation results (Figure S6), although it is known to be a challenge to simulate the optical spectra of such PAHs⁶¹,” and “Additionally, we assessed the degree of the CT character using the Lambda parameter, which varies between 0 and 1, with small values corresponding to long-range (i.e. CT) excitations and large values to short range (local) excitations⁶⁴ (see the SI for details). For the S₁ transition simulated by the TD-DFT calculation, we obtained a value of 0.7 for **4a** in acetonitrile, and larger values of 0.77–0.78 in other solvents, suggesting a partial CT contribution in acetonitrile. The overlap between holes and electrons is around 0.75 for all cases, indicating the hybridization of LE and CT characters in all solvents, in line with the experimental results.” on pages 9 in the revised manuscript as well as the simulated UV-vis absorption spectra of **4a** in the different solvents in the SI as Figure S6.

7. The authors need to do a deeper analysis of the TA and 2-D data. The assignment and explanations are not satisfactory as they do not explain the dynamics and spectra clearly enough.

Reply: We thank the referee for this valuable comment and for highlighting the importance of a deeper spectroscopic analysis. We agree that a more extensive interpretation of the transient absorption and 2D spectra could provide further insight into the excited-state dynamics. However, such an analysis would require a dedicated methodological and theoretical treatment that extends beyond the scope and focus of the present study.

In this manuscript, our goal is to report and discuss the key spectral features and dynamical trends related to the presence of a bright CT state (as detailed in the previous answer). We feel that a deeper quantitative analysis, particularly of the ultrafast kinetic components and cross-peak evolution, will be the subject of a follow-up study aimed at a specialized readership in optical spectroscopy. Nonetheless, we believe that the additional clarifications we have added to address the points 4–6 from this reviewer have made the assignments and explanations clear in the revised manuscript.

Action: Please see our changes in response to the previous points.

8. I see that the authors have done a lot of experiments but all of them do not lead to a coherent story. I suggest major revisions and resubmission.

Reply: We sincerely appreciate the referee's critical assessment, which helps us to improve the clarity and impact of our manuscript. We admit that the logic line connecting through our experiments in our previous version may not be immediately clear. Nonetheless, our work was conducted based on one central hypothesis: introducing imide functionalization via cycloaddition of large PAH and maleimide can be a powerful strategy to tailor their photophysical properties. The three parts of our study systematically tested this hypothesis: In the synthesis part, the D-A cycloaddition was established as a versatile tool to build the PAH imide derivatives; in the mechanism study part, we investigated the fundamental principles governing the reaction, especially for the reaction on the second bay position, to explain the observed reaction outcomes and predict future design; in the photophysics part, we explored how the introduction of imide groups affect the photophysical properties. Here, we directly probe how the structures, which were made possible through the previous two parts, exhibit unique excited state dynamics. Above all, the mechanism study part is not separated but provides foundational understanding for the synthesis and both of these are prerequisites for a meaningful interpretation of the photophysical properties. We have further improved the logical connections throughout the manuscript and believe that our story about the synthesis and characterizations of circumpyrene tetracarboxydiimides, using the Diels–Alder reaction and discovering the intriguing bright CT state, is sufficiently coherent and our point about the bright CT state is now clearer thanks to the valuable comments from the reviewer.

Reviewer #3 (Remarks to the Author):

The manuscript by Chen et al. describes the reaction of an extended polycyclic aromatic hydrocarbon (PAH) with maleimide derivatives through a Diels-Alder reaction localized to the two bay areas of the ovalene skeleton. This allows direct conversion of the PAH to the corresponding tetracarboxydiimide in one step without passing through the anhydride as is sometimes the case. During the synthesis, the authors note some peculiar behavior in that the mono-adduct, formed in equal yield, is unreactive towards further conversion to the diadduct. This is rationalized through computational modelling of the reaction to determine the activation energies for transformation of the plausible intermediates. Further photophysical characterization of the mono- and di-adducts provides some fundamental properties that could be of use to the community.

Reply: We sincerely thank the reviewer for his/her thoughtful review of our manuscript and pointing out the significance of our work.

1. Not surprisingly, given the expertise of the authors in this area, the research is conducted with care and the results discussed in the frame of the broader context of PAH chemistry. The real question for this reviewer is the general direction of this research: the topics covered in the introduction are noble (OFET, NLO, NIR), but it is hard to see how they can be impacted with compounds prepared in < 20% isolated yield from already complex starting materials. The small HOMO-LUMO energy gap is quite common for extended PAH, and the observation of ICT not unexpected upon introduction of acceptor groups.

Reply: We agree with the reviewer that the real application of our compounds is currently limited by their low synthetic yield. While small HOMO-LUMO energy gaps and ICT are indeed common for PAHs, the specific combination of largely extended PAHs and electron withdrawing imide groups allows us to probe the limit of charge transfer and its consequences on the excited state landscape. We believe the detailed mechanism understanding of the D-A reaction toward larger PAH imides and the new insight into the photophysical properties described in this work can be instrumental for the further development of functional PAH materials in the field. Moreover, although the synthetic yield is low at the current stage, we consider that we can potentially further improve it based on the now obtained mechanistic understanding, which is however out of the scope of the current work.

Action: We have revised the description in the manuscript to better reflect the primary contribution of our work, weakening the statements about potential applications.

2. Perhaps more useful to the community would be a comparison of the calculated vs. experimental spectroscopic properties (to assist in future computational design) and the singlet-triplet energy gap (for TADF applications). The latter is more easily calculated than measured,

but perhaps the observation of delayed fluorescence could show this.

Reply: We are grateful to the reviewer for this insightful suggestion. However, it is actually still a challenge to precisely predict the optical properties of some PAHs like circumpyrene, for which different commonly used theoretical methods can give contradicting results [e.g. see *J. Chem. Phys.* **2019**, *150*, 124302]. We would thus like to avoid the direct comparison of a single set of TD-DFT calculation results, only by one method, and the experimental absorption spectra, and use our theoretical results only for qualitative assessments. We calculated the singlet-triplet energy gaps using Gaussian 09 software at the B3LYP/6-31G(d,p) level of theory and found that the introduction of electron withdrawing imide groups decreases the energy gap from 1.11 eV for DBOV to 0.93 eV and 0.78 eV for **3a** and **4a**, respectively. Namely, the singlet energy levels of these were computed to be similar (DBOV: 2.02 eV, **3a**: 2.09 eV, **4a**: 2.01 eV) and the triplet energy levels were increased (DBOV: 0.91 eV, **3a**: 1.16 eV, **4a**: 1.23 eV). We are not familiar with the TADF applications, but these singlet-triplet energy gaps are seemingly too high to achieve the reverse intersystem crossing at reasonable temperatures. Moreover, in view of the above-mentioned theoretical challenge and lack of the experimental data regarding the triplet state, we would like to refrain from publishing these premature results and address this in our future projects. Unfortunately, our initial attempts to measure the phosphorescence of DBOV have failed, but we intend to continue our investigations related to the excited triplet state of such PAHs.

Action: In the supporting information, we have added one extra figure showing the simulated UV-vis absorption of **4a** in different solvents, involving *n*-hexane, THF, DCM, toluene and acetonitrile, as Figure S6. In addition, we have also added Table S1 to list the major electronic transitions computed for **4a** in these solvents.

3. The more interesting aspects of the transient spectroscopy will be published separately, which sort of weakens the impact of this work. It can, however, be published in its present form and without any modifications if it is of sufficient interest for the journal.

Reply: We thank the referee for this remark. We appreciate the concern that reserving part of the transient spectroscopic analysis for a separate publication could be perceived as reducing the overall impact of the present work. Our intention, however, is to maintain a clear focus here on the photophysical mechanisms and structure–property relationships that are most relevant to the broad chemistry readership.

Response to Reviewers' Comments

Reviewer #2 (Remarks to the Author):

WE thank the authors for giving a detailed point-by-point response and certainly improving the manuscript for scientific clarity. This reviewer is ok with the revisions although few minor improvements are suggested as below:

Reply: We sincerely appreciate the reviewer for their valuable and insightful comments, which greatly helped us to improve the scientific clarity of our manuscript. These two new comments raised by the reviewer have been addressed as detailed below. We hope that the revised manuscript now meets with the reviewer's standard and that the manuscript is suitable for acceptance.

1. For the REPLY to our question number 2, the authors write the text:

"As to the dehydrogenation mechanism, we believe that excess amount of maleimide play role as an oxidant in the dehydrogenation process. Indeed, in a previously reported work (*Org. Lett.*, 2003, 5, 2833), maleimide derivatives were also found to function as acceptor for hydrogen in D-A addition reaction." If this were true will the dehydrogenation reaction not work when maleimide is a LIMITING reagent without any O₂?

Reply: We thank the reviewer for this thoughtful comment. In principle, using limited amount of maleimide in the D-A addition might avoid the dehydrogenation of the D-A adduct if all the maleimide molecules are consumed as dienophile. However, whether it will work as expected also depends on the relative rate ratio of the D-A addition versus hydrogen abstraction. If the dehydrogenation process is much faster, then it will not be possible to see the hydrogenated intermediates. For *N*-hexyl and *N*-(*tert*-butylbenzyl) maleimides, we could not see any sign of hydrogenated D-A addition intermediates, probably due to this reason. However, when we used *N*-(4-*tert*-butylphenyl)maleimide for the D-A addition experiment, in addition to the expected monoadduct (**3c**), we also observed hydrogenated mono- and di-adducts by MALDI-TOF MS (see Figure R1a below; These compounds were not further characterized by ¹H NMR due to small quantity). This is reasonable considering that the electron-withdrawing ability (and oxidation ability) of this type of *N*-aryl-substituted maleimide is weaker than the other two alkyl substituted ones, so the dehydrogenation rate can be slower. Moreover, when we treated the hydrogenated intermediates with DDQ, we could clearly observe the dehydrogenation (Figure R1b).

Action: In our revised manuscript, we had written about the experimental observation of partially dehydrogenated D-A addition intermediate for the reaction with *N*-(4-*tert*-butylphenyl)maleimide on page 7. To make it clearer, we have added "This partially

dehydrogenated intermediate was not observed for the D-A addition between DBOV-Mes and **2a/b**, presumably due to the stronger oxidizing ability of these two maleimides.” after the previous sentence. The following Figure R1 has also been added in the Supporting Information as Figure S8 and referenced in the revised manuscript on page 7.

Figure R1. a) MALDI-TOF MS spectrum showing the formation of partially dehydrogenated D-A adducts for the reaction of DBOV-Mes and *N*-(*tert*-butylbenzyl) maleimide (**2c**); b) MALDI-TOF MS spectrum measured after further dehydrogenation of the intermediates in a) by DDQ using toluene as solvent ($c = 0.2$ mmol/L).

2. For the REPLY to our question number 7, the authors write the text:

"In this manuscript, our goal is to report and discuss the key spectral features and dynamical trends related to the presence of a bright CT state (as detailed in the previous answer). We feel that a deeper quantitative analysis, particularly of the ultrafast kinetic components and crosspeak evolution, will be the subject of a follow-up study aimed at a specialized readership in optical spectroscopy. Nonetheless, we believe that the additional clarifications we have added to address the points 4 – 6 from this reviewer have made the assignments and explanations clear in the revised manuscript." This reviewer is not fully satisfied by this answer since it seems that 2D data was just shown to artificially enhance the impact of the work. I suggest the authors to remove this part in all seriousness. Already CT evidences have been provided by other techniques.

Reply: We sincerely appreciate the reviewer’s frank feedback and understand the concern that including 2D electronic spectroscopy (2DES) data might appear superfluous. However, after careful consideration, we respectfully disagree with the reviewer and chose to retain the 2DES data (Figure 5e) because it provides unique spectral evidence that standard Transient Absorption (TA) cannot easily convey in a single plot.

Specifically, we utilize the 2DES map here effectively as a high-resolution "excitation-dependent pump-probe" experiment. While standard TA provides the response at a single pump

wavelength, the 2D map simultaneously visualizes the system's response across the entire excitation bandwidth. This is critical for our central claim of a "bright" Intramolecular Charge Transfer (CT) state. The 2D data clearly demonstrates that the CT state has a distinct "spectral address":

1. Excitation < 640 nm results in a diagonal Local Excitation (LE) signal.
2. Excitation > 640 nm immediately activates a distinct, broad, red-shifted feature (CT state).

This visual confirmation that the CT state can be directly and selectively accessed optically (rather than being populated solely via relaxation) is a key finding of our work.

Action: To address the reviewer's valid concern that the data was not analysed in enough depth, we have significantly expanded the description in the manuscript. We now provide a more detailed interpretation of the pump-wavelength dependence and the specific spectral features in section 2.4, ensuring the 2D data contributes substantively to the physical picture.

Reviewer #3 (Remarks to the Author):

The authors have commented on my remarks, but did not bring substantial modifications of the manuscript. While I still believe that, without the additional spectroscopic data, this work will have little impact outside the field, it can be published in its present format.

Reply: We thank the reviewer for the additional assessment. In response to the earlier remarks about the need for a more quantitative treatment of the transient absorption data, we have performed a full global analysis of the 660 nm pump–probe dataset (Figure R2). The results of this analysis (including the decay-associated spectra and extracted time constants) are now reported in the Supplementary Information (Figures S15) and are explicitly referenced in the revised manuscript text (Section 2.4).

This global analysis confirms the presence of an instrument-response-limited component, an intermediate ≈ 5 ps process, and a long-lived (>500 ps) component, fully consistent with our assignment of a bright CT-like state that is selectively accessed upon red-edge excitation. We hope that making these additional spectroscopic data and their quantitative treatment available in the Supplementary Information alleviates at least part of the reviewer's concern, while keeping the main text focused on the key physical conclusions.

Figure R2. Decay associated spectra obtained from global fitting analysis of the transient absorption spectra for **4a** pumped at 660 nm.